# Treatment of Chronic Refractory Pain by Combined Deep Brain Stimulation of the Anterior Cingulum and Sensory Thalamus (EMOPAIN Study): Rationale and Protocol of a Feasibility and Safety Study

**DOI:** 10.3390/brainsci12091116

**Published:** 2022-08-23

**Authors:** Aurélie Leplus, Michel Lanteri-Minet, Anne Donnet, Nelly Darmon, Jean Regis, Denys Fontaine

**Affiliations:** 1Fédération Hospitalo-Universitaire INOVPAIN, Centre Hospitalier Universitaire de Nice, Université Côte d’Azur, 06001 Nice, France; 2Department of Neurosurgery, Centre Hospitalier Universitaire de Nice, Université Côte d’Azur, 06001 Nice, France; 3Pain Department, Centre Hospitalier Universitaire de Nice, Université Côte d’Azur, 06001 Nice, France; 4INSERM U1107, Neuro-Dol, Auvergne University, 63001 Clermont-Ferrand, France; 5Pain Clinic, Timone Hospital, 13005 Marseille, France; 6Department of Functional and Stereotactic Neurosurgery, Assistance Publique-Hôpitaux de Marseille, Timone Hospital, 13005 Marseille, France; 7Institut Neurosciences des Systèmes, Aix-Marseille University, Institut National De La Santé Et De La Recherche Médicale, 13284 Marseille, France

**Keywords:** anterior cingulate cortex, deep brain stimulation, protocol, refractory pain

## Abstract

Background: Deep Brain Stimulation (DBS) of the sensory thalamus has been proposed for 40 years to treat medically refractory neuropathic pain, but its efficacy remains partial and unpredictable. Recent pilot studies of DBS targeting the ACC, a brain region involved in the integration of the affective, emotional, and cognitive aspects of pain, may improve patients suffering from refractory chronic pain. ACC-DBS could be complementary to thalamic DBS to treat both the sensory-discriminative and the affective components of chronic pain, but the safety of combined DBS, especially on cognition and affects, has not been studied. Methods: We propose a prospective, randomized, double-blind, and bicentric study to evaluate the feasibility and safety of bilateral ACC-DBS combined with unilateral thalamic DBS in adult patients suffering from chronic unilateral neuropathic pain, refractory to medical treatment. After a study period of six months, there is a cross-over randomized phase to compare the efficacy (evaluated by pain intensity and quality of life) and safety (evaluated by repeated neurological examination, psychiatric assessment, cognitive assessment, and assessment of affective functions) of combined ACC-thalamic DBS and thalamic DBS only, respectively. Discussion: The EMOPAIN study will show if ACC-DBS is a safe and effective therapy for patients suffering from chronic unilateral neuropathic pain, refractory to medical treatment. The design of the study will, for the first time, assess the efficacy of ACC-DBS combined with thalamic DBS in a blinded way.

## 1. Introduction

Deep brain stimulation (DBS) has been proposed since the 1970s to treat refractory chronic pain using two intracerebral targets: the sensory thalamus and the periaqueductal and periventricular grey matter [1,2]. The latter target was recommended for nociceptive, although DBS of the sensory thalamus has been rather used for refractory neuropathic pain. Thalamic DBS targeted the ventral posteromedial (VPM) and ventral posterolateral (VPL) nuclei, the site of pathological neuronal hyperactivity secondary to deafferentation [3,4]. DBS for pain was commonly practiced in the 1980s and 1990s, and its results have been reported in numerous case series (reviewed in [5,6,7]). In the case of neuropathic pain (mainly facial neuropathic pain and post plexus avulsion pain), reported thalamic-DBS efficacy varied from 40% to 60% in published studies; efficacy is usually defined as pain improvement > 50%. However, thalamic DBS has been used less since the emergence of cortical stimulation, and several studies report partial, insufficient, or short-lasting efficacy [8].

Recently, the anterior and dorsal cingulate gyrus has been proposed as a novel target for DBS for refractory pain [9]. The anterior cingulate gyrus (ACC) is located on the medial side of the frontal lobe, around the corpus callosum. It can be functionally and histologically distinguished into its dorsal part (Brodman’s area 24), located above the corpus callosum (CC), its rostral part (area 32), anterior to the CC, and its subgenual part (area 25), located below the knee of the CC. The dorsal anterior cingulate plays a functional role in motivation and affect management. The cingulate belongs to the Pain Matrix, a set of cortico-subcortical regions activated during pain processes [10]. Functional brain imaging studies suggest that the anterior cingulate is involved in the management of the cognitive, emotional, and affective components of pain (reviewed in [10,11]. In particular, the ACC is thought to be involved in the process of attributing unpleasantness or “suffering” to the experience of pain perception.

The dorsal ACC has been used as a target for the neurosurgical treatment of chronic pain, especially by performing focal lesions called cingulotomies. Case series report variable efficacy, around 50% improvement [12,13]. Patients treated by cingulotomy reported a dissociation between the persistence of the usual pain perception and a certain indifference to pain linked to the loss of perception of its unpleasant aspect. The main adverse effect of cingulotomy was apathy, although its incidence is unclear.

Therefore, chronic electrical stimulation of the ACC (DBS-ACC), a non-invasive and reversible technique, was proposed as an alternative to cingulotomy. First, in an isolated case [14], and more recently, in a series of 16 patients with refractory neuropathic pain [15], the DBS leads were implanted in the same region of the ACC that was previously targeted for lesions. In this series, DBS-ACC treatment induced a reduction of pain intensity below a VAS score of 4/10 in one-third of the patients. The overall health status (EQ-5D) and quality of life (SF-36) of the 16 treated patients improved significantly (respectively, by 20% and 7%) after ACC-DBS at a mean follow-up of 13 months. Improvement of quality of life in patients who had no change in pain intensity suggests that DBS-ACC may induce changes in the cognitive and emotional integration of pain, as if the patients had become partially indifferent to their pain, as some patients spontaneously reported [15].

DBS of the anterior cingulate (DBS-ACC) appears to be an innovative and hopeful approach for the treatment of patients with severe chronic pain resistant to all other treatments. However, these encouraging preliminary data from DBS ACC on a small number of patients need to be confirmed by a prospective study. Moreover, the ACC is involved in many cognitive and emotional processes. These functions have not been studied in patients treated with chronic ACC stimulation. Furthermore, one may fear that indifference to pain, a therapeutic effect sought and reported by ACC-DBS patients, may be associated with an emotional indifference (anhedonia) extended to other domains and/or with a loss of motivation (apathy) which may have an impact on patients’ lives. An evaluation of the cognitive, emotional, and affective consequences of DBS-ACC appears to be essential before going further in its therapeutic development.

Finally, the place of ACC-DBS in relation to thalamic-DBS remains to be clarified. In the study by Boccard et al., 7 out of 16 patients had been previously treated unsuccessfully with thalamic or periaqueductal DBS, and DBS-ACC was therefore presented as an alternative to “classic” DBS. However, these two DBS modalities have different mechanisms of action. Thalamic DBS is supposed to modulate the transmission of the nociceptive message and inhibit the pathological hyperactivity of deafferented neurons, which tends to reduce the intensity of the pain. ACC-DBS would act (under reserve of the data available) by modulating the emotional integration of pain at the level of the ACC without modifying the intensity and the perception of pain. Because of their different mechanisms of action, these two techniques could be complementary or even synergistic, acting on several dimensions of chronic pain and its perception.

To answer these questions, we propose an exploratory study evaluating the feasibility and tolerance of an approach combining ACC-DBS and thalamic-DBS in patients with severe chronic neuropathic pain for which all other therapeutic alternatives have failed. This trial will systematically investigate the possible cognitive, emotional, and affective consequences of DBS-ACC.

All patients will be treated with active and adjusted thalamic DBS for the duration of the study in order to provide a validated and potentially effective therapeutic solution. In case of poor tolerance of the thalamic stimulation, it will be stopped. The study will include a randomized phase during which ACC-DBS will be alternatively active or inactive in order to evaluate its effects in a comparative way. The lack of sensation of DBS-ACC will allow its evaluation in a double-blind phase. We will not include patients presenting with post-stroke pain in this study in order to ensure the structural integrity of the neural networks that potentially support the mechanism(s) of action of ACC-DBS. We will not include patients who respond to rTMS and, therefore, potentially respond to cortical stimulation, a technique considered less invasive than DBS.

## 2. Methods and Design

### 2.1. Study Design

The EMOPAIN study is a bicentric prospective, feasibility, and safety study to evaluate bilateral ACC-DBS combined with unilateral thalamic DBS in patients suffering from refractory chronic unilateral pain. The study design is summarized in Figure 1. The total duration of the study will be 23 months.

All patients will be treated with DBS-thalamic, activated one day after surgery and remaining active for the duration of the study in order to provide a validated and potentially effective therapeutic solution. ACC-DBS will be activated 1 month after surgery (M1), and parameter settings will be optimized during the next 3 months (M1–M4).

In order to evaluate the efficacy and safety of ACC-DBS, the study will also include a randomized phase during which DBS-ACC will be alternatively active or inactive. The lack of sensation induced by DBS-ACC will allow its double-blind evaluation. Randomization will take place 4 months after the surgery (Figure 1). The aim of the randomized phase is to compare the combination of DBS-ACC and DBS-thalamic with DBS-thalamic alone, with the patient as his own control, in two 3-month periods organized in a cross-over design. The thalamic stimulation will be left on throughout this phase. The patient will be randomized (centralized randomization) between a DBS-ACC sequence on (“On”) and a DBS-ACC sequence off (“Off”), these parameters being set by the neurosurgeon. Patients and evaluating neurologists will be blinded for the results of randomization. The two sequences will be separated by a two-week wash-out period to avoid a residual effect. Evaluation data will be collected by the neurologist at the end of each sequence (M7 and M10).

### 2.2. Endpoints

The feasibility of ACC-DBS will be evaluated by the proportion of patients undergoing with success the process of surgical intervention, chronic stimulation, and evaluation without serious adverse events. Safety will be evaluated by repeated general and neurological examination, psychiatric assessment, and assessment of cognitive and affective functioning. The global cognitive assessment includes mini mental status (MMSE), the French version of the Free and Cued Selective Reminding Test (FCSRT), the GREFEX battery, including the Trail Making Test (TMT) [16], the Stroop test, the six elements test, the Brixton test, the double task, the modified card sorting test (MCST), and verbal fluencies [17], and the Digit Span and Digit Symbol-coding (WAIS IV) [18]. Assessment of affective functions will be performed using the Hospital Anxiety and Depression (HAD) scale, the Lille Apathy rating scale (LARS), the revised version of the “Reading the mind in the eyes” test [19], the Facial Expressions of Emotion—Stimuli and Tests (FEEST) scale [20], and the Multidimensional Assessment of Thymic States [21].

DBS efficacy will be evaluated using pain intensity on the Visual Analogic Scale (VAS), Brief Pain Inventory, McGill Pain Questionnaire, Patients’ Global Impression of Change (PGIC), and quality of life improvement (EQ-5D-3L health questionnaire).

Assessments will be performed 1 month before surgery and 1, 4, 7, 10, and 22 months after.

### 2.3. Patients

We will include 10 adult patients (age 18–70 years old) suffering from chronic (duration > 1 year) unilateral neuropathic pain, severe (VAS score > 6/10), with high emotional impact (HAD sub-scores > 10), considered resistant to medication specific to neuropathic pain at sufficient doses and durations (including at least antiepileptics and antidepressants), and not sufficiently improved by rTMS and relevant surgical solutions. Patients who respond to rTMS will not be included as they may potentially respond to cortical stimulation, a technique considered to be less invasive than DBS.

Non-inclusion criteria will be cognitive impairment (MMSE score < 24), DSMIV axis I psychiatric disorder, and contra-indication to surgery, DBS, anesthesia, or MRI.

Considering the pioneering nature of this study and the number of patients required in similar pilot studies exploring the effects of DBS in emerging indications, it seems to us that 10 patients would be sufficient to achieve the objectives of this project.

### 2.4. Ethicals Considerations

The design of the study has been approved by the Ethical Medical Committee (“Comité de Protection de Personnes Sud Meditteranée”, 2017). Informed consent will be obtained from all patients before inclusion. There will be no financial reward for the patients.

### 2.5. Interventions

The day before the operation, a morphological brain MRI, including diffusion tensor imaging (DTI) sequence, will be performed on a 3 Tesla MRI in order to obtain high-quality brain images, guaranteeing quality surgical targeting.

The operation consists of the unilateral lead implantation in the sensory thalamus contralateral to the pain side, combined with bilateral lead implantation in the dorsal anterior cingulate gyrus. These electrodes will then be connected by subcutaneous cables to two abdominal or pectoral subcutaneous generators: one for the thalamic stimulation, the other for the cingulate stimulation.

The stereotactic implantation technique will be similar to the technique already used for DBS in movement disorders and other conditions. Similar to what may be used in Parkinson’s disease, this operation will be performed under local anesthesia and will include a phase of intraoperative micro-electrode recording and “test” stimulation in order to optimize the implantation of the thalamic lead within the sensory thalamic nuclei.

### 2.6. Definition and Implantation of the Thalamic Target

Under local anesthesia, a stereotaxic frame will be fixed on the patient’s head. The target will be determined by so-called “indirect” targeting, i.e., in reference to the mean coordinates of the target in relation to the anatomical landmarks constituted by the anterior (CA) and posterior (CP) commissures identified on the MRI (T1 3D SPGR sequence after injection of Gadolinium) performed under stereotactic conditions. The coordinates of the sensory thalamus are y = 2–3 mm anterior to CP; at the bi-commissural plane (z = 0). Laterally, the x-coordinate will differ depending on the topography of the pain due to the somatotopy of the ventral posterior lateral (VPL) and medial (VPM) thalamic nuclei. An x = 12–13 mm (relative to CA-CP) will be used in cases of facial pain and x = 14–15 mm in cases of upper limb pain. The entry point will be frontal, defined to avoid the trajectory crossing cortical vessels and ventricle.

The optimal position of the electrode will be refined by intraoperative electro-physiological exploration using micro-electrode recording and test stimulation [22]. Indeed, tactile or proprioceptive stimulation in the body region where the pain is located classically induces changes in the discharge pattern of the thalamic neurons corresponding somatotopically to this region. This phenomenon will be used to localize the thalamic region corresponding to the painful area and in which the lead will be implanted. Intraoperative stimulation (as well as chronic stimulation) of the VPM-VPL induces pleasant paresthesia in the contralateral body region corresponding to the stimulated thalamic area. Intraoperative test stimulation will be used to check the absence of adverse effects related to stimulation (muscle contractions, dysarthria, hot flushes, diplopia, etc.).

Once the optimal lead implantation site is identified, an octopolar electrode (Abbott Infinity 8 contact-directional lead, Chicago, IL, USA) will be implanted. The use of a directional lead will help to avoid possible side effects by preferentially steering the electrical stimulation in the optimal direction. The lead location will be checked by intra-operative X-rays images, according to the technique routinely used in each center.

### 2.7. Definition and Implantation of the Anterior Cingulate Target

The identification of the cingulate target will be done by so-called “direct targeting”, i.e., by visualization of the target on stereotactic MRI, according to the technique and location proposed by Boccard et al. [8,15]. The target is located in the dorsal anterior cingulate, approximately 20 mm posterior to the projection of the anterior tip of the frontal horn of the lateral ventricle. The trajectory will be chosen to ensure that the top three contacts of the lead are located at the junction between grey matter (cingulate gyrus) and white matter (cingulate fasciculus) and the deepest contact in the corpus callosum. The entry point will be frontal, determined to avoid the cortical vessels. The two ACC-DBS leads will be implanted bilaterally and symmetrically, without an electro-physiological exploration phase.

### 2.8. Connections to the Generators

Between 0 and 5 days after the implantation of the electrodes, a second operation will be performed, under general anesthesia, to connect the 3 electrodes to 2 dual-channel stimulators (Infinity 5 and 7, Abbott, Chicago, IL, USA) placed in 2 subcutaneous pectoral, sub clavicular, or abdominal pockets (according to the patient’s choice). The thalamic electrode will be connected to one stimulator and the cingulate electrodes to the other stimulator so that each target can be stimulated independently with different parameters.

## 3. Defining the Stimulation Amplitudes

### 3.1. Thalamic Stimulation

During the postoperative hospitalization, thalamic stimulation will be started using the contact closest to the optimal target, with the following stimulation parameters: frequency 20–100 Hz, pulse width 150 ms, amplitude 1–4 V. These stimulation parameters will be adapted to ensure that the paresthesias induced by stimulation of the sensory thalamus are pleasant and felt in the painful region. The thalamic stimulation will be used with the optimal parameters (the most effective on pain and the best tolerated) during the whole study.

### 3.2. Cingulate Stimulation

Stimulation of the ACC does not induce any perceptible feeling. The stimulation parameters used in the present study will be based on those used by Boccard et al. [15]: 4 to 6.5 Volts, frequency 130 Hz, pulse width 450 ms. To avoid a “kindling” effect and the risk of epilepsy, the chronic stimulation will be cyclic, alternating a 5-min “On” phase and a 10-min “Off” phase. The amplitude will be increased progressively, and the stimulating contacts possibly modified, depending on the therapeutic or adverse effects observed during the period between M1 and M4. The parameters found to be the most effective and best tolerated will be used for the randomized phase.

### 3.3. Statistical Analysis

Given the preliminary nature of this study, the analysis of the data concerning the efficacy of this approach will be essentially descriptive and will concern the evolution of the scores of the pain scales and neuropsychological scales over time. The results will be expressed in the form of individual graphical curves showing the evolution of these scores. These curves will make it possible to compare this evolution before and after stimulation.

The tolerance and efficacy of the combined DBS-ACC and thalamic DBS will be evaluated by comparing the postoperative scores at M22 with the preoperative scores (“baseline”). The tolerance and efficacy of DBS-ACC will be assessed by comparing the scores of each of the randomized sequences DBS-ACC “On” and DBS-ACC “Off”. Due to the small number of subjects in this study, statistical analysis will be based on non-parametric tests.

## 4. Discussion

In this article, we describe the rationale and design of an ongoing prospective, double-blind, parallel-group multicenter RCT to evaluate the feasibility and safety of bilateral ACC-DBS combined with unilateral thalamic DBS in patients suffering from chronic unilateral pain, refractory to medical treatment.

Until now, the only available data on the possible efficacy of ACC-DBS comes from small, open studies, which are insufficient to prove effectiveness and classify ACC-DBS as a regular therapy. On another hand, thalamic DBS has been proposed for years to treat intractable pain. We aim to evaluate the efficacy of combined ACC-DBS and thalamic-DBS because preliminary data suggested that these two approaches may be complementary to improving chronic pain patients. However, the design of our study might allow obtaining a clue to the respective effects of each DBS target, as thalamic DBS will be used alone for one month, and ACC-DBS will be evaluated in a cross-over On/Off trial. The efficacy of these single and combined DBS effects will be evaluated not only using pain intensity measures (VAS, Brief Pain Inventory, McGill Pain Questionnaire) but also using emotional pain impact and quality of life assessments (SF36, EQ5D).

Before going further in its therapeutic development, it appears to be essential to evaluate the cognitive, emotional, and affective consequences of DBS-ACC.

We will include only patients suffering from chronic neuropathic pain. In a meta-analysis, Bittar et al. [5] showed that DBS of the sensory thalamus was more efficient on nociceptive pain than neuropathic pain. However, nociceptive pain may be more easily controlled by pharmacological approaches than neuropathic pain. In their study evaluating ACC-DBS in 16 patients, Boccard et al. [15] included various etiologies of chronic pain (brachial plexus injury, post-stroke pain, and spinal pathology, including traumatism and FBSS), but efficacy did not differ across etiologies. ACC-DBS is supposed to modulate the affective component of pain, whatever the cause of pain.

However, it is possible that certain pain etiologies may respond better to ACC-DBS than others. In our study, the pathophysiology of pain will be relatively homogenous; only patients with chronic refractory neuropathic pain will be included. We will not include patients with post-stroke pain in this study to ensure the structural integrity of the neural networks that potentially support the DBS-ACC mechanism(s) of action.

We expect DBS-ACC to be successful in chronic and refractory neuropathic pain and to be well tolerated. We expect that its effect on the emotional and affective components of pain will not be accompanied by undesirable consequences on the overall cognitive, affective, and emotional functioning of patients. This would confirm the interest in DBS-ACC as a potential alternative or complement to DBS-thalamic and would offer refractory chronic pain patients a new therapeutic option.

This study will also allow the collecting of data for the construction of a larger study.

## Figures and Tables

**Figure 1 brainsci-12-01116-f001:**
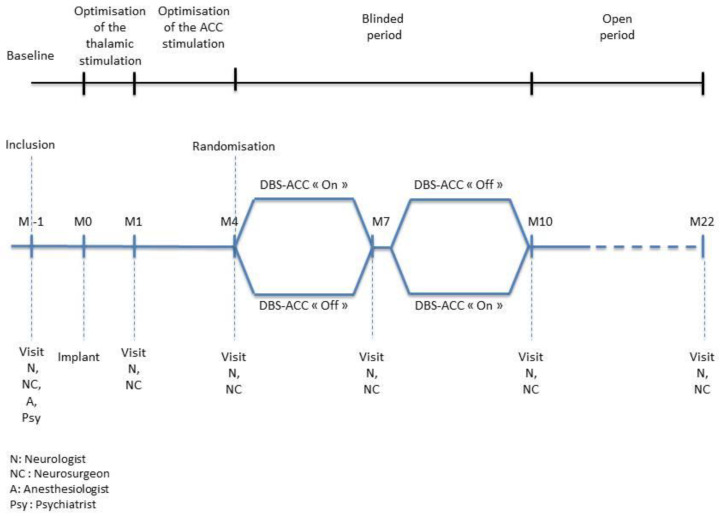
Study design. The study design consists of a 1-month pre-treatment evaluation phase, a phase of thalamic stimulation alone (1 month), and then thalamic and cingulate stimulation (3 months M1 to M4). A 6.5-month cross-over randomized phase will compare DBS-ACC combined with thalamic DBS to thalamic DBS alone (M4 to M7 and M7 to M10). Then a 12-month open phase will evaluate the long-term effect of combined DBS-ACC and thalamic DBS stimulation (M10 to M22).

## Data Availability

Not applicable.

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
