# Peer review of "Treatment of Chronic Refractory Pain by Combined Deep Brain Stimulation of the Anterior Cingulum and Sensory Thalamus (EMOPAIN Study): Rationale and Protocol of a Feasibility and Safety Study"

_brainsci, 2022, doi:10.3390/brainsci12091116_

Round 1

Reviewer 1 Report

This article presents the Study Protocol for a prospective, randomised, double blind and bicentric study to evaluate the feasibility and safety of bilateral dorsal anterior cingulate cortex DBS combined with unilateral thalamic DBS in adult patients suffering from chronic unilateral neuropathic pain. The point made was that this study protocol will assess whether “ACC-DBS is a safe and an effective therapy for patients suffering from chronic unilateral neuropathic pain” as an add-on treatment to unilateral thalamic DBS. The subject matter is novel. The authors do a good job curating the literature to conclude that the “design of the study will, for the first time, assess efficacy of ACC-DBS in a blinded way”. Some suggestions are listed in more detail below.

Comments for the authors:

1. In the abstract, what do the authors mean by the ACC “bilateral DBS of the dorsal ACC […] has been successfully proposed to treat few patients”. I would remove the word successfully and rephrase this since the dACC remains an investigational target due to its efficacy remaining partial and unpredictable, similarly to thalamic DBS.

2. Whilst the study protocol is well-presented, it is important to make it clearer that it specifically addresses the question of whether additional bilateral ACC stimulation on top of unilateral thalamic stimulation may confer benefit (or be safe/feasible) – as opposed to assessing unilateral or bilateral ACC stimulation alone. Hence, the authors cannot say that this study will show if ACC-DBS is a safe and an effective therapy or assess efficacy of ACC-DBS alone, as erroneously written in the Discussion of the abstract. It should be said more precisely that it will, for the first time, specifically assess ACC-DBS as an add-on to simultaneous thalamic DBS.

3. The authors should comment on how the pathophysiology of pain may also be heterogenous, which may account for variation of efficacy and targets used in DBS for chronic pain. hence, the manuscript could benefit from a comment from the authors on how patient selection make impact this protocol.

4. In Figure 1, Randomisation is marked at M7, however, text says it “will take place 4 months after surgery”. Please clarify and correct it.

5. In Methods and design, the authors state that “it seems to us 168 that 10 patients”, however, they do not clearly state in the protocol whether they have an a priori sample size.

6. More detail about the wash-out phase needs to be provided. How are investigators going to proceed if the two-week wash-out period is not enough for patients to go back to baseline levels, for example, in case of long-lasting effects of the months of stimulation? Will the DBS-thalamic remain UNCHANGED during randomization AND wash-out AND crossover? The authors mention it will stay on but should provide more details on that.

Minor comments:

1. In the introduction, it would flow better if the authors connected the paragraph mentioning ACC lesions to the mention to cingulotomies. The paragraph about the pain matrix seems a bit disjointed.

2. There are a few typos, formatting errors, and missing punctuation throughout the text (e.g., lines 48,  155, 210, 214, etc)

Author Response

Point 1: 1. In the abstract, what do the authors mean by the ACC “bilateral DBS of the dorsal ACC […] has been successfully proposed to treat few patients”. I would remove the word successfully and rephrase this since the dACC remains an investigational target due to its efficacy remaining partial and unpredictable, similarly to thalamic DBS.

Response 1: The sentence has been rephrased : Recent pilot studies of DBS targeting the ACC a brain region involved in the integration of the affective, emotional and cognitive aspects of pain may improve patients suffering from refractory chronic pain.   

Point 2: 2. Whilst the study protocol is well-presented, it is important to make it clearer that it specifically addresses the question of whether additional bilateral ACC stimulation on top of unilateral thalamic stimulation may confer benefit (or be safe/feasible) – as opposed to assessing unilateral or bilateral ACC stimulation alone. Hence, the authors cannot say that this study will show if ACC-DBS is a safe and an effective therapy or assess efficacy of ACC-DBS alone, as erroneously written in the Discussion of the abstract. It should be said more precisely that it will, for the first time, specifically assess ACC-DBS as an add-on to simultaneous thalamic DBS.

Response 2: We thank the reviewer for this relevant comment. The sentence in the abstract discussion has been revised : “The design of the study will, for the first time, assess efficacy of ACC-DBS combined with thalamic DBS in a blinded way.”

We also added at the end of the introducion:

“All patients will be treated with active and adjusted thalamic DBS for the duration of the study in order to provide a validated and potentially effective therapeutic solution. In case of poor tolerance of the thalamic stimulation, it will be stopped. The study will include a randomised phase during which ACC-DBS will be alternatively active or inactive, in order to evaluate its effects in a comparative way. The absence of perceptible effect of DBS-ACC will allow its evaluation in a double-blind phase. “

And in the discussion, as required by the reviewer:

“On another hand, thalamic DBS has been proposed for years to treat intractable pain. We aim to evaluate the efficacy of combined ACC-DBS and thalamic-DBS, because preliminary data suggested that these two approaches may be complementary to improve chronic pain patients. However the design of our study might allow to get a clue of the respective effects of each DBS target, as thalamic DBS will be used alone during one month, and ACC-DBS will be evaluated in a cross-over On/Off trial. The efficacy of these single and combined DBS effect will be evaluated not only using pain intensity measures (VAS, Brief Pain Inventory, McGill Pain Questionnaire) but also using emotional pain impact and quality of life assessments (SF36, EQ5D).”

Point 3. The authors should comment on how the pathophysiology of pain may also be heterogenous, which may account for variation of efficacy and targets used in DBS for chronic pain. hence, the manuscript could benefit from a comment from the authors on how patient selection make impact this protocol.

We have developed this issue and added a paragraph in the discussion as follows:

We will include only patients suffering from chronic neuropathic pain. In a meta-analysis, Bittar et al showed that DBS of the sensory thalamus was more efficient on nociceptive pain than neuropathic pain. However nociceptive pain may be more easily controlled by pharmacological approaches than neuropathic pain. In their study evaluating ACC-DBS in 16 patients, Boccard et al had included various eatiologies of chronic pain (brachial plexus injury, post stroke pain, spional pathology: traumtism and FBSS) but efficacy did not differ across aetiologies. ACC DBS is supposed to modulate the affective component of pain whatever the cause of pain.

However it is possible that certain pain etiologies may respond better to ACC DBS than others. In our study the pathophysoilogy of pain will be relatively homogenous: only patient with chronic refractory neuropathic pain will be included. We will not include patients with post-stroke pain in this study to ensure the structural integrity of the neural networks that potentially support the DBS-ACC mechanism(s) of action.

Point 4. In Figure 1, Randomisation is marked at M7, however, text says it “will take place 4 months after surgery”. Please clarify and correct it.

Response 4: The correction has been done in figure 1 M7 is replaced by M4. M10 by M7 and M25 by M22 in accordance with the text below.

The design consists in : a one-month pre-treatment evaluation phase, a phase of optimisation of the thalamic stimulation parameters (1 month) and then cingulate stimulation (3 months M1 to M4). A 6.5-month ran-domised phase comparing DBS-ACC combined with thalamic DBS to thalamic DBS alone M4 to M7 and M7 to M10, and then a 12-month open phase of combined DBS-ACC and thalamic DBS stimulation, M10 to M22.

Point 5. In Methods and design, the authors state that “it seems to us that 10 patients”, however, they do not clearly state in the protocol whether they have an a priori sample size.

The sample size is 10 patients.

We added in the Methods chapter : “We will include 10 adult patients (age 18-70 years old) suffering …”

Point 6. More detail about the wash-out phase needs to be provided. How are investigators going to proceed if the two-week wash-out period is not enough for patients to go back to baseline levels, for example, in case of long-lasting effects of the months of stimulation? Will the DBS-thalamic remain UNCHANGED during randomization AND wash-out AND crossover? The authors mention it will stay on but should provide more details on that.

All patients will be treated with active and adjusted thalamic DBS for the duration of the study, including during the wash out period, in order to provide a validated and potentially effective therapeutic solution. In case of poor tolerance of the thalamic stimulation, it will be stopped.

A wash-out phase of 2 weeks is planned during which ACC-DBS will be stopped. A carry-over effect cannot be excluded. However as the patients will be evaluated at the end of each 3-month period of the randomised phase, it is unlikely that a carry-over effect will be still present at that time.

Minor comments:

  1. In the introduction, it would flow better if the authors connected the paragraph mentioning ACC lesions to the mention to cingulotomies. The paragraph about the pain matrix seems a bit disjointed.

To follow the reviewer comment, the sentence about lesions has been removed (“Damage to the ACC results in an overall decrease in interest, a decrease in motivation and activity leading to apathy, and a blunting of affect”) for a better comprehension.

  1. There are a few typos, formatting errors, and missing punctuation throughout the text (e.g., lines 48, 155, 210, 214, etc)

Typos and formatting erros has been corrected.

Reviewer 2 Report

DBS of the anterior cingulate (DBS-ACC) appears to be an innovative and hopeful approach to the treatment of patients with severe chronic pain resistant to all other treatments. An evaluation of the cognitive, emotional and affective consequences of DBS-ACC is essential before going further in its therapeutic development. The place of ACC-DBS in relation to thalamic-DBS remains to be clarified.

 To this purpose the authors performed an exploratory study by which they evaluate the feasibility and tolerance of an approach combining ACC-DBS and thalamic-DBS, in patients with severe chronic neuropathic pain for which all other therapeutic alternatives have failed.

Some remarks:

Methods and design:

- the duration of the study is not specified. Please add. - it’s not clear how many patients were included in the study. At lines 168-69 you wrote:“ it seems to us 168 that 10 patients would be sufficient to achieve the objectives of this project”. This means you included 10 patients in your study? Please clarify - Fig 1: the explanations are brief. Please give more details. Discussion: -at Methods and design, point 2.2. Endpoints, you wrote about the methods used to evaluate the feasibility and safety.  Why didn't you present and  discuss your results? -in my oppinion the discussions are missing. Only the study protocol is presented. Some discussions should be added.
